# MRI-Guided Targeted and Systematic Prostate Biopsies as Prognostic Indicators for Prostate Cancer Treatment Decisions

**DOI:** 10.3390/cancers15153915

**Published:** 2023-08-01

**Authors:** Furat Abd Ali, Karl-Dietrich Sievert, Michel Eisenblaetter, Barbara Titze, Torsten Hansen, Peter J. Barth, Ulf Titze

**Affiliations:** 1Bielefeld University, Medical School and University Medical Center OWL, Klinikum Lippe Detmold, Department of Urology, 32756 Detmold, Germany; furat.abdali@klinikum-lippe.de (F.A.A.); karl-dietrich.sievert@klinikum-lippe.de (K.-D.S.); 2Bielefeld University, Medical School and University Medical Center OWL, Klinikum Lippe Detmold, Department of Diagnostic and Interventional Radiology, 32756 Detmold, Germany; michel.eisenblaetter@uni-bielefeld.de; 3Bielefeld University, Medical School and University Medical Center OWL, Department of Pathology, Klinikum Lippe Detmold, 32756 Detmold, Germany; barbara.titze@klinikum-lippe.de; 4MVZ for Histology, Cytology and Molecular Diagnostics Trier GmbH, 54296 Trier, Germany; t.hansen@patho-trier.de; 5University of Münster, Gerhard-Domagk-Institute of Pathology, Münster University Hospital, 48149 Münster, Germany; peter.barth@ukmuenster.de

**Keywords:** prostate cancer, MRI-guided targeted biopsies, clinically significant prostate cancer

## Abstract

**Simple Summary:**

The term “clinically relevant prostate carcinoma” (csPCa) is used to differentiate the types of prostate carcinoma (PCa) that can lead to the death of the affected patient from those tumor types that usually do not. Targeted biopsies (TBx) of visible lesions in an MRI of the prostate has increased the detection of csPCa. The aim of this prospective study is to assess the extent to which the collection of TBx alone was sufficient for the correct diagnosis of therapy-relevant PCa and how often the results of systematic biopsies (SBx) were crucial for the treatment decisions. Depending on the definition of a csPCa, 80–90% of therapy-relevant tumors were detected with TBx alone.

**Abstract:**

The standard procedure for the diagnosis of prostate carcinoma involves the collection of 10–12 systematic biopsies (SBx) from both lobes. MRI-guided targeted biopsies (TBx) from suspicious foci increase the detection rates of clinically significant (cs) PCa. We investigated the extent to which the results of the TBx predicted the tumor board treatment decisions. SBx and TBx were acquired from 150 patients. Risk stratifications and recommendations for interventional therapy (prostatectomy and radiotherapy) or active surveillance were established by interdisciplinary tumor boards. We analyzed how often TBx alone were enough to correctly classify the tumors as well as to indicate interventional therapy and how often the findings of SBx were crucial for therapy decisions. A total of 28/39 (72%) favorable risk tumors were detected in TBx, of which 11/26 (42%) very-low-risk tumors were not detected and 8/13 (62%) low-risk tumors were undergraded. A total of 36/44 (82%) intermediate-risk PCa were present in TBx, of which 4 (9%) were underdiagnosed as a favorable risk tumor. A total of 12/13 (92%) high-risk carcinomas were detected and correctly grouped in TBx. The majority of csPCa were identified by the sampling of TBx alone. The tumor size was underestimated in a proportion of ISUP grade 1 tumors. Systematic biopsy sampling is therefore indicated for the next AS follow-up in these cases.

## 1. Introduction

Prostate cancer (PCa) is the second most frequent cancer and the fifth leading cause of cancer death among men in 2020, having caused more than 375,000 deaths worldwide in this period [1]. Prostate-specific antigen (PSA) screening significantly reduces the risk of death from prostate cancer [2] but incurs the risk of overdiagnosis and thus unnecessary therapy complications [3,4]. Blood and urine biomarkers have been developed to assess the individual risk profile prior to biopsy collection. Magnetic resonance imaging (MRI) has gained support as a triage test in the diagnosis of prostate carcinoma because it enables a risk classification of visible lesions by assessing their diffusion and perfusion patterns [5] and permits the targeting of biopsy needles to individual lesions.

Prostate biopsy is still the gold standard for the diagnosis of PCa. The standard procedure provides the systematic collection of 10–12 biopsies (SBx) from the apex to the base from both lobes [6]. The utility of targeted biopsies of MRI-visible lesions (TBx) has been intensely investigated. MRI findings are significant predictors of adverse pathology features [7] as MRI-visible lesions are enriched with molecular features for aggressive tumors [8]. MRI-TBx thus increase the detection rates of clinically relevant prostate cancer, whereas combining targeted and systematic biopsies offers the best chances of detecting all cancers [9].

Currently, national and international guidelines recommend a combined sampling of both SBx and TBx of MRI-visible or palpable lesions with the goal of detecting all tumors. However, recent studies indicate a paradigm shift toward a more selective detection of high-risk cancers requiring therapy and, at the same time, a more restrained strategy toward less aggressive tumors. In the GÖTEBORG-2 trial, it was shown that the samplings of TBx alone reduced the detection rate of insignificant carcinomas. However, only higher grade PCa (ISUP grade ≥ 2) were defined as therapy-relevant tumors in these studies, irrespective of other clinical and serological parameters.

In this study, we analyzed the impact of TBx and SBx on risk stratification and treatment decisions when other clinically relevant factors usually available at the time of biopsy collection are considered. In addition to tumor grading, tumor sizes determined by biopsy, serum PSA levels, patient age, and family history were considered. Primary endpoint was the level of agreement between the therapy recommendations based on the results of the TBx alone, with the final tumor board decision, which included the results of SBx. Secondary endpoint was the analysis of how often SBx were necessary to decide in favor of interventional therapy and on which criteria these decisions were based. Tertiary endpoint was used to determine the differences between the cohorts of biopsy-naïve men, men with a prior negative biopsy, and men with a known PCa under active surveillance (AS).

## 2. Materials and Methods

### 2.1. Patients and MRI Image Data

A total of 150 patients aged between 42 and 86 years (mean 65.5 ± 8.7) were included in this prospective observational study. The Participants were admitted as regular inpatients for elective prostate biopsy due to suspicion of a PCa or as follow-up biopsy for a known PCa under AS. Family history, most recent PSA level, and a recent MRI were available for each patient.

This study was based on an inhomogeneous MR data set since 82/150 (55%) patients brought MR imaging from external examiners (a total of 15 practices that provided MRI data for 1 to 28 patients). A total of 68/150 (45%) patients received MRI imaging in the radiology department of the Klinikum Lippe. All images were examined by experienced radiology specialists and were classified according to the Prostate Imaging Reporting and Data System (PIRADS) [10]. 

The external MRI image data were included in the in-house Picture Archiving and Communication System (PACS) but were not re-examined by colleagues from our Department of Radiology for legal reasons. However, as part of contouring for biopsy, the images were critically re-evaluated by experienced clinical colleagues from the Department of Urology.

### 2.2. Biopsy Procedure

The MRI scans provided detailed information about the size, shape, and location of any suspicious areas within the prostate gland. These images were fused with real-time ultrasound images taken during the biopsy procedure using specialized software that aligned the two imaging modalities (Ultrasound System BK-5000, BK medical, Burlington, MA, USA; BioJet^TM^ MRI/Ultrasound Fusion System, Medical Targeting Technologies GmbH, Barum, Germany). The urologist precisely targeted and took biopsies (Fully Automatic Reusable Biopsy System DeltaCut, PAJUNK GmbH, Geisingen, Germany) from the suspicious areas identified on the MRI scan.

Lesions suspicious of a tumor were assigned to the sectors of a systematic biopsy procedure (Figure 1). Depending on their size and location, one or more TBx were acquired from the tumor-suspected regions. In addition, SBx were taken from each of the remaining sectors of the prostate, avoiding the visible lesions.

### 2.3. Histological Examination

The biopsies were transferred to the Institute of Pathology immediately after collection. The specimens were formalin-fixed and processed according to a standardized procedure for formalin-fixed and paraffin-processed (FFPE) material. Hematoxylin–eosin (HE)-stained histological slides were examined within two working days by specialized pathologists with several years of experience in uropathology (TH, UT). Immunohistochemical combination stains for P504S and p40 were used for the classification of difficult cases. The reporting was performed according to current guidelines [11] and included the diagnosis of PCa and the graduation as Gleason score (GS) and ISUP group. The absolute sizes of the tumor infiltrates (in mm), as well as their relative sizes (in %), were reported for each biopsy. Furthermore, the findings included information about inflammatory and reactive lesions as possible causes for PSA elevations. All histological specimens underwent secondary evaluation by an external uropathologist (PJB) as part of the study.

### 2.4. Tumor Board Decisions

Risk stratification of tumor patients was performed according to the NCCN Clinical Practice Guidelines in Oncology [12]. The classification includes four risk categories: very-low-risk group (ISUP-grade 1, PSA ≤ 10, <3 biopsies involved, and <50% infiltration), low-risk group (ISUP-grade 1, PSA ≤ 10, and tumor in 3 or more biopsies/>50% infiltration), intermediate-risk group (ISUP-grades 2 + 3 and/or PSA > 10–20), and a high-risk group (ISUP-grades 4 + 5 and/or PSA > 20).

According to the national guidelines [13], staging examinations were carried out in the PCa patients of the intermediate- and high-risk groups, and the indication for individualized interventional therapy (radical prostatectomy (RPE), radiotherapy, or chemotherapy) was established. AS was indicated for patients in the very-low-risk group. The guidelines recommended interventional therapy for patients in the low-risk group. Age, family history, and age were also taken into account for the individual therapy decisions.

### 2.5. Study Design

In combination with serum PSA levels, risk stratifications were performed for both TBx and SBx results. The risk groups determined for TBx and SBx were compared with the final risk stratification of the tumor board. The sensitivity for tumor detection was determined for both types of biopsies and analyzed depending on the risk groups. The frequency of TBx correctly predicting the final risk stratification was analyzed. Finally, it was analyzed to what extent the TBx could distinguish between tumors of the favorable risk groups (low and very low risk) and the intermediate/high-risk groups.

### 2.6. Statistical Analysis

For the statistical evaluation, the risk groups were transferred to an ordinal scale (0—no tumor, 1—very-low-risk group, 2—low-risk group, 3—intermediate-risk group, and 4—high-risk group). For the detection of so-called clinically significant carcinomas (intermediate- and high-risk groups) in the TBx, the sensitivity, specificity, and the positive/negative predictive values were determined. The level of agreement with the tumor board decisions was analyzed using Cohen’s kappa [14] and interpreted according to the Landis/Koch categories [15].

## 3. Results

### 3.1. Patient Cohorts

A total of 100/150 (67%) men presented with a suspected PCa for the first time for biopsy (biopsy-naïve men). A total of 32/150 (21%) men with a persistent suspicion of PCa presented again for a prostate biopsy after no tumor could be detected in a previous biopsy procedures (men with prior negative biopsies). The smallest cohort consisted of 18/150 (12%) men with a known PCa under AS for a follow-up biopsy. Age and PSA values were evenly distributed in the groups (Table 1). The mean PSA value was 9.7 ± 8.6 ng/mL (range 1–61 ng/mL).

### 3.2. MRI Findings and Biopsy Acquisition

In the MRI images of the 150 patients, 215 tumor-suspect lesions were identified (1.4 ± 0.7 lesions per patient, range 1–4 lesions). A total of 21 patients of the total collective (14%) had tumor-suspect lesions in the anterior zones (TZ a) exclusively, which could also be reliably biopsied. A total of 44/150 (29%) patients presented with PIRADS 5 lesions; 83/150 (55%) patients showed PIRADS 4 in their scans; 18/150 (12%) patients were grouped as PIRADS 3; and 3/150 (2%) patients had PIRADS 2 lesions. In 2/150 (1%) patients, the PIRADS groups were documented inconsistently.

Depending on the sizes of the visible lesions, between one and eight MRI-fused biopsies (TBx) were obtained from each identifiable lesion (mean 3.3 ± 1.2). A total of 720 TBx (mean 4.8 ± 1.9 per patient) and 1588 systematic biopsies (SBx) (mean 10.6 ± 3.4 per patient) were collected and analyzed. 

### 3.3. Biopsy Results

A PCa was present in 96/150 (64%) patients. A total of 56/96 (58%) patients showed GS6 (ISUP grade 1) carcinoma in their biopsies. A GS3+4 carcinoma (ISUP grade 2) cancer was present in the specimens of 33/96 (34%) men. A total of 2/96 (2%) patients were diagnosed with a GS4+3 PCa (ISUP grade 3), and another 2/96 (2%) of men showed GS4+4 cancer (ISUP grade 4). A total of 3/96 (3%) men presented with GS4+5 or GS5+5 cancer (ISUP grade 5) in their biopsies.

A total of 13/21 (62%) lesions located in the anterior parts of the prostate proved to be a carcinoma, of which 7 were GS ≥ 7 tumors. A total of 10/13 (77%) tumors were seen in the TBx and 12/13 (92%) in the SBx (intersection 9/13, 69%). A total of 6/7 (85%) GS ≥ 7 were detected in the TBx.

### 3.4. Risk Stratification of the Patients and Tumor Board Decisions

In a synopsis of the biopsy results and the PSA values, 39/96 (41%) patients were assigned to the favorable group. Of these, 26/96 (27%) patients fulfilled the criteria of a very-low-risk constellation, and 13/96 (14%) belonged to the low-risk group. A total of 44/96 (46%) patients were assigned to the intermediate-risk group, and 13/96 (14%) patients had a high-risk profile. Altogether, 70/96 (73%) of patients had the indication (intermediate- or high-risk groups) or a potential indication (low-risk group) for interventional therapy. The synopsis of the MRI findings showed both an increase in the detected carcinomas and higher rates of clinically significant tumors requiring therapy with the increasing PIRADS groups (Figure 2). 

A total of 20/26 (77%) the patients in the very-low-risk group remained under active surveillance. Five of the patients under AS (17%) nevertheless opted for interventional therapy in view of their comparatively young age (60.6 ± 8.7 years, range 47–69 years). One other patient presented an ISUP-upgrade to a high-risk situation due to an external preliminary biopsy result. A total of 9/13 (69%) patients in the low-risk group had the indication for interventional therapy, whereas 4/13 (31%) of these patients opted for AS. All 44 patients of the intermediate-risk group had the indication for interventional therapy (7 patients chose radiation, and 37 patients voted for radical prostatectomy). Furthermore, all 13 patients of the high-risk group qualified for curative therapy after the clinical staging examinations (9 patients chose RPE, 3 opted for radiation, and 1 patient had the indication for chemotherapy). 

### 3.5. Tumor Detection in TBx and SBx

When comparing TBx and SBx (Figure 3), there was a slightly higher cancer detection rate for TBx. A total of 52/96 (54%) tumors were recorded in both TBx and SBx. A total of 24 tumors were exclusively represented in TBx (24 + 52 = 76, sensitivity 79%), and 20/96 tumors were not detected in TBx. Of these, 18/20 tumors were detected only in the SBx (18 + 52 = 70, sensitivity 73%). In the cohort of AS patients, no tumor infiltrates were detectable in either TBx or SBx in two cases. The tumors missed in TBx were 11 very-low-risk group cancers, 8 tumors from the intermediate-risk group, and 1 tumor from the high-risk group.

There were clear differences in tumor detection in the individual risk groups. Very-low-risk group: TBx detected 15/26 (58%) cases, whereas SBx detected 11/26 (42%) patients. A total of 2/26 (8%) cases with a known PCa under AS were missed in both TBx and SBx. A total of 22/26 (85%) cases were present in TBx or SBx alone. Only 2/26 (8%) cases were seen in TBx and SBx together. In summary, the majority of these tumors were diagnosed in either the TBx or the SBx, and about half of these cases would have remained undetected with TBx only.

Low-risk-carcinoma group: All tumors (13/13, 100%) of this group were visible in TBx and in SBx, but only 5/13 (38%) were correctly grouped using TBx due to an infiltration grade > 50% in the biopsies. In contrast, 8/13 (62%) of the cases were undergraded as very-low-risk tumors (infiltration grades < 50%) and could only be correctly grouped with the information from SBx. In 3/8 cases, SBx delivered infiltration grades > 50%, and in 5/8 cases TBx and SBx together revealed >2 biopsy cores involved. In summary, all the tumors in this group were detected using TBx but, in the majority, were underestimated in size.

Intermediate-risk group: Comparable detection rates were also found in this group. Tumor detection and correct assignment were achieved using TBx in 32/44 (73%) cases and in 33/44 (75%) cases using SBx. A total of 8/44 (18%) cases were each detected with only one of the biopsy strategies (and were thus missed in TBx). A total of 4/44 (9%) cases were wrongly assigned to the favorable risk groups with TBx alone as higher ISUP grades were only seen SBx. In summary, approximately every fifth tumor in this group was overlooked in the TBx, and every tenth case was underestimated with regard to its grading.

High-risk group: This group showed higher cancer detection rates for TBx. A total of 12/13 (92%) cases were detected and correctly assigned with TBx. In contrast, 10/13 of these cases were recorded in the SBx, of which 1 was underestimated and therefore incorrectly assigned to the intermediate-risk group.

In summary, TBx and SBx showed comparable tumor detection rates. TBx revealed a slightly higher proportion of tumors in the intermediate- and high-risk groups. Tumors of the low-risk group were also reliably detected. However, there was insufficient discriminatory power against tumors of the very-low-risk group, which were also frequently missed in TBx.

### 3.6. Detection Rates of Clinically Significant Cancer in TBx

The definition of clinically significant prostate carcinomas (csPCa) is still a matter of controversy (Table 2). Recent studies set the threshold for csPCa in GS7 tumors, whereas GS6 tumors were considered insignificant regardless of their PSA levels. Applying this criterion to our data, 40/96 (42%) of the tumors were classified as clinically significant. On the other hand, 3 cases from the high-risk group (GS6 PCa, PSA > 20 ng/mL) and 14 cases from the intermediate-risk group (GS6 PCa, PSA 10–20 ng/mL) were assessed as not being clinically significant.

Of 40 cases with GS7 PCa or higher, 36/40 (90%) tumors were detected in the TBx, and 31/40 (78%) were correctly classified as clinically significant. A total of 4/40 (10%) cases were missed, and 5/40 (13%) cases were wrongly assigned to favorable risk groups.

If cases in the intermediate- and high-risk groups were defined as clinically significant, the above-mentioned patients with GS6 carcinomas and PSA values of >10 and >20 ng/mL, respectively, were also included (57/96 cases, 59%). A total of 48/57 (84%) of these cases were detected in TBx. A total of 9/57 (16%) of these cases were missed, and 4/57 (7%) of patients were underrated and mistakenly assigned to favorable risk groups. A total of 44/57 (77%) of these cases were correctly classified as csPCa.

Local guidelines [13] defined tumors of the low-, intermediate-, and high-risk groups as clinically significant, while only cases in the very-low-risk category (PSA < 10 ng/mL, GS6 PCa in <50% of biopsies and in ≤2 cores involved) were seen as clinically insignificant, and AS was recommended. In our collective, 70/96 (73%) tumors were accordingly classified as csPCa in the tumor board, of which 61/70 (87%) cases were detected in the TBx. A total of 10/70 (14%) cases were incorrectly assigned to the very-low-risk group using TBx.

In summary, 51/70 (73%) of tumors were correctly classified as csPCa with the analysis of TBx alone. The highest discriminatory power was achieved when csPCa were defined as GS7 carcinomas or higher. When GS6 carcinomas were included in the definition of csPCa, slightly lower detection rates were achieved, and a proportion of cases were underestimated.

### 3.7. Differences in the Cohorts

Of 100 biopsy-naïve men, 65 patients had prostate carcinoma detectable in the biopsies. There were 17 (26%) patients assigned to the very-low-risk group, 8 (12%) patients presented tumors in the low-risk group, 31 (48%) patients had a tumor in the intermediate-risk group, and 9 (14%) patients presented a high-risk carcinoma. A total of 54/65 (84%) tumors were detected in TBx, and 49/65 (75%) were correctly grouped. Six tumors of the very-low-risk group and one tumor from the high-risk group were missed in the TBx. Three tumors of the low-risk group were underrated as very-low-risk tumors, and two tumors of the intermediate-risk group were mistakenly assigned to favorable risk groups.

Men with prior negative biopsies were less likely to have a PCa overall. Tumor infiltrates were detectable in the biopsies of 13/32 (41%) patients. A total of 4/13 (31%) patients had tumors from the very-low-risk group, 1/13 (12%) patients showed a low-risk tumor, and 4/13 (31%) patients each showed tumors in the intermediate- and high-risk groups. In this cohort, SBx had a greater impact on tumor detection than in the other groups. A total of 7/13 (53%) tumors were recorded in TBx, and 5/13 (38%) cases were correctly classified. Three tumors in each of the very-low-risk and intermediate-risk groups were not detected with TBx. One low-risk and intermediate-risk tumor each were underestimated.

Of 18 men with a known PCa under AS, 5/18 (28%) continued to show a very-low-risk constellation, 4/18 (22%) patients showed a low-risk carcinoma, and in 9/18 (50%) patients, there was an upgrade to the intermediate-risk group. There were 8/18 (44%) patients who remained in AS, and 10/18 (56%) chose interventional therapy (7/10 RPE, 3/10 Radiation). Carcinoma infiltrates were present in 15/18 (83%) in the TBx and were correctly classified in 10/18 (56%). Two tumors in the very-low-risk group and one tumor in the intermediate-risk group were missed in the TBx. Four tumors of the low-risk group and one tumor of the intermediate-risk group could only be classified by the additional information of the SBx.

### 3.8. Overall Diagnostic Performance of the TBx Alone

Depending on their definitions, csPCa were detected with a sensitivity between 70–90% and a specificity of 100%. Due to false negative findings and undergraduation, negative predictive values resulted between 70–90%. Calculating with the overall patients examined, we found substantial levels of agreement of the analysis of the TBx with the final tumor board for the detection of csPCa requiring therapy. In our study, the MR pathway proved to be a suitable tool to selectively identify a high proportion of carcinomas requiring therapy. However, this would have meant overlooking or underestimating a proportion of carcinomas that were considered to be in need of therapy according to currently valid guidelines.

## 4. Discussion

PCa has been long known to be a morphologically, genetically, and clinically heterogeneous disease. The tumors often harbor multiple morphologically and clonally distinct foci. Clinical courses vary from indolent localized tumors in elderly men to widespread metastatic disease in younger individuals. Clinical management of a PCa is a difficult tradeoff between early detection and treatment of metastatic disease on the one hand and avoidance of unnecessary procedures with treatment complications on the other.

PSA screening led to a reduction in PCa-induced deaths but incurred the risk of overdiagnosis and the complications of curative treatment [3]. There is therefore an urgent need for a diagnostic system that can detect high-risk tumors as selectively as possible. An important milestone for filtering out patients who have low-risk PCa is the multiparametric MRI of the prostate and its capability for targeted biopsy of suspicious lesions. The advantages of a more specific detection of clinically significant PCa in TBx and the risks of their underdiagnosis by omitting SBx are currently the subject of intense debate [16,17,18]. The most important finding of the GÖTEBORG-2 trial was that the omission of SBx halved the detection rate of clinically insignificant tumors but missed one in five clinically significant cancers [19]. The tumors found only in the systematic biopsies were smaller and from the intermediate-risk group (GS7a).

The term “clinically relevant prostate carcinoma” (csPCa) is used to differentiate a PCa that can lead to the death of the affected patient from tumor types that usually do not. The overtreatment of insignificant cancers, which account for a large proportion of PCa [20], has been cited as a disadvantage of PSA screening [21]. Although pathologic findings are used to determine csPCa, its definition consists of a combination of factors of tumor biology and individual factors of the patient, especially family history, actual and biological age, comorbidities, and life expectancy. There is a broad consensus that high-risk PCa is significant for all men (except when life expectancy is limited). Recent papers have defined csPCa differently, commonly using ISUP grade 2 and above, which demonstrates a need for further discussion and an overall consensus [17,18,22,23,24,25].

In conventional risk stratifications, which are based on the histologic results of systematic biopsies, and PSA levels, GS7 carcinomas are assigned to at least the intermediate-risk category and are classified as clinically significant and thus in need of therapy [26]. In large studies, csPCa was defined as CS7 cancers or higher, whereas CS6 carcinomas were classified as not significant in the biopsies, regardless of their size [19,22,27]. Large studies of prostatectomy specimens demonstrated that GS6 cancers rarely show biochemical recurrences or extraprostatic extensions and never metastasize into lymph nodes [28,29]. These studies led to calls for GS6 carcinomas to no longer be referred to as a “cancer” [30]. Defining GS ≤ 6 PCa as an insignificant cancer does not mean it should be ignored, but rather appropriately observed, and AS is becoming the preferred treatment option [31].

It is important to note that GS6 cancer in a biopsy can incur a risk of developing metastasis and disease-specific death, due to the possibility of the under-sampling of a higher-grade component. If the tumor sizes > 8 mm in the biopsies, the likelihood is 90% to be upgraded from GS6 to GS7 in the prostatectomy specimens [32]. Molecular studies show that tumor size is an independent factor in the accumulation of genomic risk factors [33]. Furthermore, molecular studies have shown that even small foci can metastasize [34]. Till today, there is no agreement on the maximum number of systematic cores that can be involved with cancer or the maximum percentage of core involvement, although there was recognition that an extensive disease on MRI should exclude men from AS [11]. Based on autopsy studies, GS6 carcinomas >0.5 mL were defined as clinically significant [35]. Only smaller tumors were assigned to the “very low risk” category, which resulted in the current valid criteria for AS (GS6, <50% infiltration, and ≤2 biopsy cores involved). The authors had already discussed that this definition was probably too stringent and aimed to avoid consequent undertreatment.

The choice of biopsy strategy depends on the type of tumors to be detected. The current guidelines recommend the combined collection of TBx and SBx with the aim to detect all carcinomas [11,13]. Our data are consistent with previous studies [27] and support the concept of the MRI pathway with TBx alone from a clinical–practical perspective. As a limitation, it should be mentioned that our study was based on a patient population with inhomogeneous MRI data, which, however, reflects the reality in a clinical routine.

Cancer detection rates in TBx were comparable to those in SBx, with TBx being more sensitive to intermediate- and high-grade tumors. TBx showed the highest discriminatory power for the detection of GS7 PCa or higher (sensitivity 90% and correct assignment in 80%). However, almost every third GS6 tumor was not recorded in TBx, of those, 20% were classified as clinically significant (intermediate risk or even high risk) according to the currently valid guidelines. This restriction seems tolerable since rapid tumor progression is not to be expected in these cases. Multiple autopsy data show that 36–51% of men harbored an undiagnosed PCa that did not limit their life expectancy and—if detected—would have been considered in need of therapy [36]. Recent studies have shown that it is also reasonable to offer delayed treatment to some patients with an intermediate-risk PCa [37]. However, there are currently no valid criteria for an adequate patient selection [38].

The strategy of isolated MRI-guided TBx appears to be appropriate for the initial biopsy of both men who have never undergone prostate biopsy and those under AS. The omission of additional SBx would reduce the rate of septic complications and local inflammation, which should have a beneficial effect on any subsequent surgical procedures. Men with previous negative biopsies should always receive MRI imaging in case of persistent suspicions of a tumor. As in previous studies, we also found lower cancer detection rates in this cohort both absolutely and specifically in TBx. Only slightly more than one-third of csPCA could be detected with TBx alone in this cohort, so a combined sampling of TBx and SBx is recommended for this patient population [39], which we fully support. Promising results from recent studies suggest that the implementation of AI-augmented lesion detection and PIRADS scoring in MRI as well as robotic-assisted biopsy sampling techniques will provide further steps towards a more specific detection of csPCa [40].

## 5. Conclusions

The acquisition of targeted biopsies from prostate lesions visible on MRI allowed the detection and correct assignment of the vast majority of prostate carcinomas requiring therapy. In doing so, it was accepted that a proportion of GS6 carcinomas would be underestimated or even overlooked. Additional systematic biopsies defined 10–30% of significant tumors (by their size or upgrade) depending on the cohort but also led to a higher detection of non-significant carcinomas. Our results support the concept of the MRI pathway of diagnosis of prostate carcinoma with the acquisition of targeted biopsies alone for the timely detection of tumors in need of therapy on one hand and to avoid unnecessary diagnostic and therapeutic procedures on the other.

## Figures and Tables

**Figure 1 cancers-15-03915-f001:**
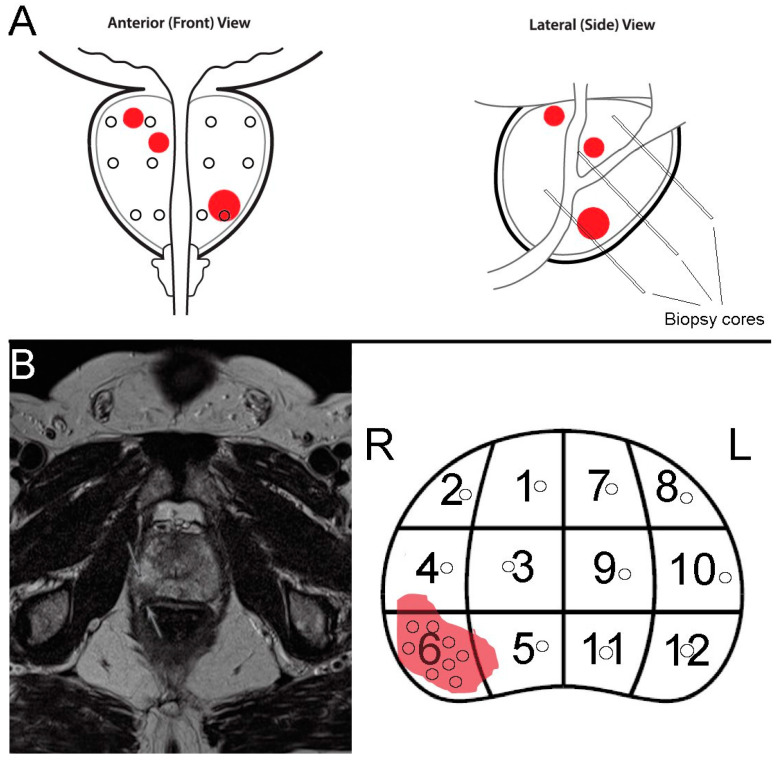
Biopsy protocols. (**A**): Protocol for systematic biopsy (SBx) sampling—a total of 12 biopsies were taken evenly from apical, central, and basal sections of both prostate lobes. In this scheme, the largest and most dorsally located tumor site (red lesions) was captured in one biopsy, while the smaller and more anteriorly located tumor sites were not encountered. (**B**): Multimodal MRI showed a tumor-suspicious lesion in the right dorsolateral prostate apex with concomitant diffusion distortion (PIRADS 4 lesion). The remaining tissue had signs of nodular hypertrophy and increased or decreased signal intensity. For this study, one or more biopsies (black circles) were taken specifically from the target region (TBx), depending on the size of the lesions on MRI. Additionally, one biopsy was taken from each of the remaining sites according to the standard procedure (numbered zones) for systematic biopsies (SBx), avoiding the tumor-suspicious lesion.

**Figure 2 cancers-15-03915-f002:**
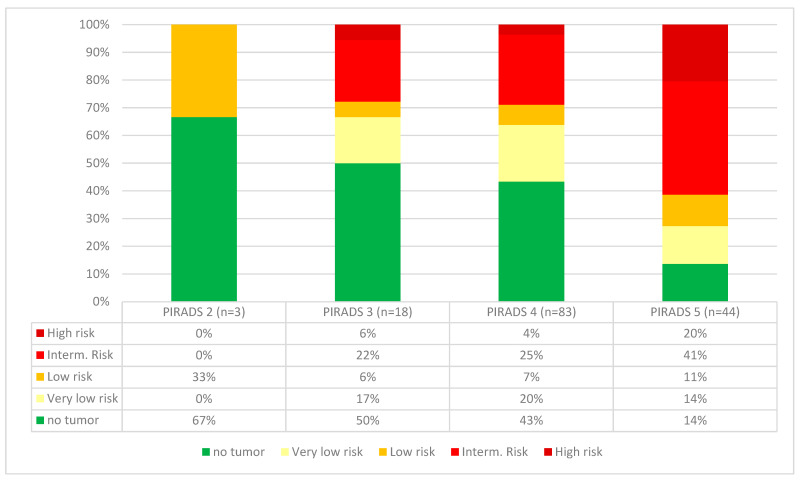
Cancer detection rates of the PIRADS groups. In higher PIRADS groups, both the rate of tumors detected and the proportion of diseases requiring therapy increased.

**Figure 3 cancers-15-03915-f003:**
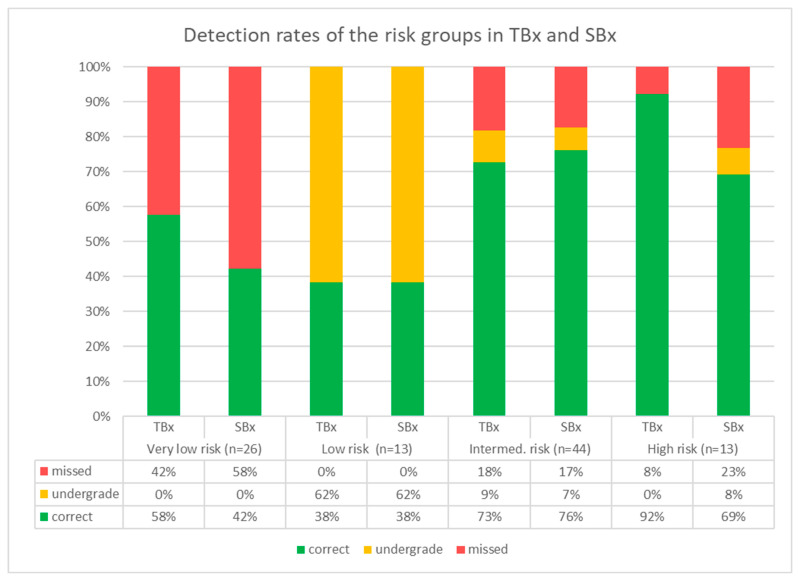
Cancer detection rates of the risk groups in targeted biopsies (Tbx) and systematic biopsies (SBx). TBx achieved high detection rates and correct stratifications in intermediate- and high-risk cancer groups. Low-grade cancers were impeccably detected but often undergraded. Nearly half of the VL-risk cancers were missed in TBx. TBX and SBx showed comparable detection rates in all risk groups, but TBx showed slightly better results in high-grade cancers.

**Table 1 cancers-15-03915-t001:** Impact of TBx and SBx on therapy decisions.

Patient Cohorts	Biopsy-Naïve Men		Negative Previous Biopsy		Active Surveillance		All Patients	
	*n*	%	*n*	%	*n*	%	*n*	%
Patients	100	67%	32	21%	18 **	12%	150	100%
Age (Ø, ±)	66.7	8.9	69.0	8.2	69.4	8.0	67.5	8.7
PSA (Ø, ±)	9.9	9.6	10.6	6.6	7.2	4.6	9.7	8.6
**Overall Biopsy results**								
No Tumor	35	35%	19	59%	2 **	11%	54	36%
Tumor	65	65%	13	41%	16	89%	96 **	64%
Sensitivity for TBx	54	83%	7	54%	15	83%	76	79%
Tumor in TBx only	17	26%	3	23%	4	22%	24	25%
Sensitivity for SBx	48	74%	10	77%	12	67%	70	73%
Tumor in SBx only	11	17%	6	46%	1	6%	18	19%
Tumor in TBx and SBx	37	57%	4	31%	11	61%	52	54%
**Risk groups**								
Very-low-risk	17	26%	4	31%	5	28%	26	27%
Low-risk	8	12%	1	8%	4	22%	13	14%
Intermediate-risk	31	48%	4	31%	9	50%	44	46%
High-risk	9	14%	4	31%	0	0%	13	14%
**TBx results**								
Cancer detection	54	83%	7	54%	15	83%	76	79%
Missed tumors	11	17%	6	46%	3	17%	20	21%
Very-low-risk	6	9%	3	23%	2	11%	11	11%
Low-risk	0	0%	0	0%	0	0%	0	0%
Intermediate-risk	4	6%	3	23%	1	6%	8	8%
High-risk	1	2%	0	0%	0	0%	1	1%
Underrated tumors	5	8%	2	15%	5	28%	12	13%
Low-risk	3	5%	1	8%	4	22%	8	8%
Intermediate-risk	2	3%	1	8%	1	6%	4	4%
High-risk	0	0%	0	0%	0	0%	0	0%
Correct assignment	49	75%	5	38%	10	56%	64	67%

** The cohort of patients under active surveillance included 18 men with a known PCa in the very-low-risk group. However, a carcinoma was detectable in only 16 patients. In total, tumor infiltrates were recorded in only 94 patients, although the entire collective included 96 tumor patients.

**Table 2 cancers-15-03915-t002:** Statistical comparison of the TBx predictions and the final tumor board decisions.

Tumor Board Stratification		Risk Stratification in TBx			
	No Tumor	Very Low Risk	Low Risk	Interm. Risk	High Risk
**No tumor (*n* = 54)**	54	0	0	0	0
**Very low risk (*n* = 26)**	11	15	0	0	0
**Low risk (*n* = 13)**	0	8	5	0	0
**Interm. risk (*n* = 44)**	8	2	2	32	0
**High risk (*n* = 13)**	1	0	0	0	12
**Definition of csPCa**		**All PCa**	**Low–High Risk**	**Interm. + High Risk**	**ISUP > 1**
N (cases)		96	70	57	40
Sensitivity		79%	73%	77%	78%
Specificity		100%	100%	100%	100%
Positive pred. value		100%	100%	100%	100%
Negative pred. value		73%	81%	88%	92%
p0		0.867	0.873	0.913	0.940
pe		0.502	0.511	0.550	0.637
Cohen’s kappa		0.73	0.74	0.81	0.83

The upper part of the table compares NCCN risk groups based on TBx with the final tumor board in an error matrix. The overlooked and underestimated cases of each risk group can be easily identified in this presentation. In the lower part of the table, the detection rate of clinically significant PCa is analyzed depending on possible definitions (all tumors/low-risk to high-risk/intermediate- and high-risk/GS7+: GLEASON score 7 or higher).

## Data Availability

The data are not publicly available due to privacy restrictions. The data presented in this study are available upon reasonable request from the corresponding author.

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
