# Peer review of "MRI-Guided Targeted and Systematic Prostate Biopsies as Prognostic Indicators for Prostate Cancer Treatment Decisions"

_cancers, 2023, doi:10.3390/cancers15153915_

Round 1
Reviewer 1 Report
This study has merit and clinical significance, and builds on some earlier work by members of the authorship.
The methods are appropriate, reported in sufficient detail, and realistically interpreted. Clearly, the objectives are important due to the paucity of reliable biomarkers that are available for PCa diagnosis and prognosis and that often lead to either over- or under-diagnosis and consequently inappropriate treatment decisions.
The assertion that MRI-guided data, in spite of the imprecise definition of csPCa, is a valuable tool alone by TBx analysis in some patient risk categories that will be valuable to both physicians and pathologists regarding diagnosis and determining treatment regimens is evidence based and the treatment of results is sound.
The manuscript would benefit from some proofreading e.g. page 3 line 105 should read 'GLEASON' score but is otherwise well-presented and I recommend publication.
Very minor attention to grammatical matters of English and a few typographical amendments.
Author Response
This study has merit and clinical significance, and builds on some earlier work by members of the authorship.
The methods are appropriate, reported in sufficient detail, and realistically interpreted. Clearly, the objectives are important due to the paucity of reliable biomarkers that are available for PCa diagnosis and prognosis and that often lead to either over- or under-diagnosis and consequently inappropriate treatment decisions.
The assertion that MRI-guided data, in spite of the imprecise definition of csPCa, is a valuable tool alone by TBx analysis in some patient risk categories that will be valuable to both physicians and pathologists regarding diagnosis and determining treatment regimens is evidence based and the treatment of results is sound.
The manuscript would benefit from some proofreading e.g. page 3 line 105 should read 'GLEASON' score but is otherwise well-presented and I recommend publication.
*****
We thank you for the favorable review of the manuscript. The text was revised and proofread several times.
*****
Reviewer 2 Report
This is a very interesting paper which tried to identify TBx and SBx in deciding prostate cancer treatment option. There are two main problems I found in this paper:
First, is MRI reliable or not, the second reading is needed to check if the false negative rate is high. This is because SBx has avoided the most MRI suspicious sites in biopsy and still detected 70 patients (nearly half of the cohort) with cancer.
Secondly, the purpose of this study is to evaluate both biopsies, but the reference for this study is still pathology results (both biopsies) with PSA and size of lesions, etc. can you play and judge at the same time?
Here are some comments for details:
1. Line 74: is this a prospective study or retrospective study?
2. Line 81: MRI scans details are missing here; how many radiologists were involved? Are they in the author list? Their work experience? Was there any secondary evaluation done?
3. Line 82: what MRI images were imported in the ultrasound machine? Please list the brand of ultrasound machine and specify what software was used in your fusion biopsy. Please list the urologist’s experience and is she/he in the author list?
4. Line 90-97: in Figure 1, was TBx only taken from zone 6? Were Zone 4/5 ignored and only took SBx? If MRI located lesion is in two zones or more, how did you take the biopsy sample?
5. How about the large but anteriorly located lesion? Were they targeted and what is the percentage in all TBx?
6. Line 102: please specify pathologists’ working experience.
7. Line 149: Table 1: in overall biopsy results for all patients, in total there are 96 patients with tumor confirmed by either TBx or SBx and it was mentioned that TBx and SBx were not taking the same location, which means the highest suspicious areas in MRI have been missed by SBx. Under this assumption, there are still 70 patients (nearly half of the cohort) confirmed with cancer, so I would like to know how many cancers have been missed by MRI? At the same time, 50 patients (76+70-96) were both detected by TBx and SBx, have both biopsies taken the same lesion or different lesions. It needs to be explained clearly in Fig 1.
8. Line 195: the combination of TBx and SBx is 96 in Table 1 (overall biopsy results for all patients), where is 94 from?
9. Line 266: there are 65/100 biopsy naïve men are positive, they are both confirmed by TBx and SBx biopsy, are there any cancers missed by both biopsies?
Author Response
This is a very interesting paper which tried to identify TBx and SBx in deciding prostate cancer treatment option. There are two main problems I found in this paper:
*****
We thank Reviewer 2 for the detailed review of the manuscript and appreciate the effort put into this work. We are grateful for the helpful comments and hope to be able to answer the points adequately.
*****
First, is MRI reliable or not, the second reading is needed to check if the false negative rate is high. This is because SBx has avoided the most MRI suspicious sites in biopsy and still detected 70 patients (nearly half of the cohort) with cancer.
*****
Answer: The approach of our study was not to question and verify the performance of the MRI. We regard this as a given fact in knowledge of the literature. In our opinion, MR imaging represents a fundamental cornerstone for the diagnosis of clinically relevant prostate carcinomas (MRI-pathway), which is increasingly practiced in clinical routine. The aim of the present study was to assess the extent to which the isolateted acquisition of TBx reflected the risk stratification of a complete biopsy procedure (TBx + SBx).
This prospective study included patients from routine clinical practice who underwent biopsy procedures at our prostate cancer center. Therefore, this study is based on an inhomogeneous MR data set. 82/150 (55%) patients brought MR imaging from 15 external examiners. The image data were transferred to the in-house PACS system, but were not re-examined by colleagues from our Department of Radiology for legal reasons. However, as part of contouring procedure for MRI/US-fusion, the images were critically reevaluated by clinical colleagues from the Department of Urology.
Since prostate cancer is a multifocal process in the majority of cases (80% of patients, Haffner et al), high detection rates were to be expected in the SBx. In the majority of cases, tumor infiltrates were detected in both TBx and SBx, so that further evaluation had to deal with this intersection. Of 96 tumors, 52 (54%) were recorded in both TBx and SBx. 24 tumors were exclusively represented in TBx (24+52= 76, sensitivity 79%). 18 tumors were detected only in SBx (18+52=70, sensitivity 73%).
The result is in agreement with previous studies on detection rates of TBx and SBx (Mischinger, frontiers in Surgery, 2022).
*****
Secondly, the purpose of this study is to evaluate both biopsies, but the reference for this study is still pathology results (both biopsies) with PSA and size of lesions, etc. can you play and judge at the same time?
*****
Answer: We do not see the decision-making power in any single position. The serum PSA is a given fact, available before the beginning of the biopsy procedure. The pathologist examines the biopsies regardless of PSA and MRI imaging. The biopsy of a patient with a PIRADS2 lesion is analyzed in the same manner as that of a patient with a PIRADS5 lesion. The histological report determines whether a tumor is present, provides the available information on tumor size (the length of the tumor infiltrate in mm and the percentage of the core involved or expressed as the number of biopsies affected), and assigns a malignancy grade to the tumor. Risk stratification is performed on the basis of the above-mentioned variables in the interdisciplinary tumor board. The serum PSA in particular often plays a decisive role.
*****
Here are some comments for details:
- Line 74: is this a prospective study or retrospective study?
*****
Answer: This study is based on a prospective design. Added in the text.
*****
- Line 81: MRI scans details are missing here; how many radiologists were involved? Are they in the author list? Their work experience? Was there any secondary evaluation done?
*****
Answer: The external MRIs (for 82/150 patients) were obtained and assessed by a total of 15 radiology practices. There were five cooperation partners (ALRA Detmold; Gemeinschaftspraxis Lippstadt; DIRANUK Bielefeld; röntgen Paderborn and radiox), which produced images for 4-28 patients (together 69/82, 86%). The remaining 13 patients presented MRI images from a total of 9 other radiology practices. These are experienced colleagues who have been cooperating with the prostate carcinoma centers in the region for several years. All contributing practices were included in the acknowledgements.
*****
- Line 82: what MRI images were imported in the ultrasound machine? Please list the brand of ultrasound machine and specify what software was used in your fusion biopsy. Please list the urologist’s experience and is she/he in the author list?
*****
Answer: Procedures for MRI/US fusion and biopsy collection described in the text, naming the instruments used. Text adapted. The physicians were experienced urologists with several years of professional experience (Prof. Dr. K.-D. Sievert and F. Abd Ali) and are represented in the list of authors.
*****
- Line 90-97: in Figure 1, was TBx only taken from zone 6? Were Zone 4/5 ignored and only took SBx? If MRI located lesion is in two zones or more, how did you take the biopsy sample?
*****
Answer: The figure has been revised. The principle of biopsy acquisition was explained more deeply. If the foci were large lesions that extended over two or more zones, they were not biopsied as SBx. The SBx avoided the foci visible on MRI and were taken as an add-on to the TBx.
*****
- How about the large but anteriorly located lesion? Were they targeted and what is the percentage in all TBx?
*****
Answer: In this specific case, only the dorso-apical lesion was classified as tumor suspicious (PIRADS 4) due to an associated diffusion disorder, whereas the anterior lesion was evaluated as a low-signal area in the context of prostatic hyperplasia. In fact, the prostatectomy specimen of this patient was available. Hyperplastic stromal nodules indeed dominated in the anterior sections. This patient showed a multifocal tumor with small foci of 1-2 mm in size in the anterior zones, which, however, may not have been relevant for imaging on MRI.
21 patients of the total collective (14%) had tumor-suspect lesions in the anterior zones (TZ a), which could also be reliably biopsied. 13 of these foci proved to be carcinoma, of which 7 were csPCa (GS7+). 10/13 tumors were met in the TBx, 12/13 (92%) in the SBx (intersection 9/13, 69%). 6/7 (85%) csPSA were detected in the TBx.
****
- Line 102: please specify pathologists’ working experience.
*****
Answer: The pathologists have several years of experience in the field of uropathology and have worked at several prostate cancer centers. Text revised.
*****
- Line 149: Table 1: in overall biopsy resultsfor all patients, in total there are 96 patients with tumor confirmed by either TBx or SBx and it was mentioned that TBx and SBx were not taking the same location, which means the highest suspicious areas in MRI have been missed by SBx. Under this assumption, there are still 70 patients (nearly half of the cohort) confirmed with cancer, so I would like to know how many cancers have been missed by MRI? At the same time, 50 patients (76+70-96) were both detected by TBx and SBx, have both biopsies taken the same lesion or different lesions. It needs to be explained clearly in Fig 1.
*****
Answer: As discussed above, most tumors were detected in both TBx and SBx. The table and text have been revised accordingly. 20/96 tumors were not detected in TBx. Among them were 11 tumors of the very low risk group, 2 of which were not represented in the SBx either (see answer to question 8). Furthermore, 8 tumors of the intermediate risk group and one tumor of the high risk group were not detected in the TBx, but only in the SBx.
*****
- Line 195: the combination of TBx and SBx is 96 in Table 1 (overall biopsy results for all patients), where is 94 from?
*****
The cohort of AS patients made the evaluation more complicated. This group included 18 patients with known PCa of the very low risk group. However, carcinoma infiltrates were detectable in only 16 of these patients. In total, tumor infiltrates were recorded in only 94 patients, although the entire collective included 96 tumor patients. The table was revised and commented accordingly.
*****
- Line 266: there are 65/100 biopsy naïve men are positive, they are both confirmed by TBx and SBx biopsy, are there any cancers missed by both biopsies?
*****
Answer: Table 1 now contains the necessary information. In this cohort, tumor was recorded in both TBx and SBx in 37/65 patients, in 17 patients only in the TBx, and in 11 patients only in the SBx. In this cohort, 4 intermediate risk tumors and 1 high risk tumor were missed in the TBx.
It must remain unanswered how many tumors were missed in all biopsies (TBx and SBx), as this is a prospective study. These patients are currently considered tumor-free and undergo further follow-up according to guideline recommendations.
*****